# Overexpression of Murine *Rnaset2* in a Colon Syngeneic Mouse Carcinoma Model Leads to Rebalance of Intra-Tumor M1/M2 Macrophage Ratio, Activation of T Cells, Delayed Tumor Growth, and Rejection

**DOI:** 10.3390/cancers12030717

**Published:** 2020-03-18

**Authors:** Annarosaria De Vito, Paola Orecchia, Enrica Balza, Daniele Reverberi, Debora Scaldaferri, Roberto Taramelli, Douglas M. Noonan, Francesco Acquati, Lorenzo Mortara

**Affiliations:** 1Human Genetics Laboratory, Department of Biotechnology and Life Sciences, University of Insubria, 21100 Varese, Italy; a.devito@uninsubria.it (A.D.V.); d.scaldaferri1@uninsubria.it (D.S.); roberto.taramelli@uninsubria.it (R.T.); francesco.acquati@uninsubria.it (F.A.); 2Molecular Pathology Unit, IRCCS Ospedale Policlinico San Martino, 16132 Genova, Italydaniele.reverberi@hsanmartino.it (D.R.); 3Cell Biology Unit, IRCSS Ospedale Policlinico San Martino, 16132 Genova, Italy; enricabalza54@gmail.com; 4Immunology and General Pathology Laboratory, Department of Biotechnology and Life Sciences, University of Insubria, 21100 Varese, Italy; douglas.noonan@uninsubria.it; 5Scientific and Technology Pole, IRCCS MultiMedica, 20138 Milan, Italy

**Keywords:** RNASET2, C51 colon carcinoma, M1/M2 macrophage ratio, TNFα-producing M1 macrophages, immune T cell responses

## Abstract

Human RNASET2 acts as a powerful oncosuppressor protein in in vivo xenograft-based murine models of human cancer. Secretion of RNASET2 in the tumor microenvironment seems involved in tumor suppression, following recruitment of M1-polarized macrophages. Here, we report a murine *Rnaset2*-based syngeneic in vivo assay. BALB/c mice were injected with parental, empty vector-transfected or murine *Rnaset2*-overexpressing mouse C51 or TS/A syngeneic cells and tumor growth pattern and immune cells distribution in tumor mass were investigated. Compared to control cells, mouse *Rnaset2*-expressing C51 cells showed strong delayed tumor growth. CD86^+^ M1 macrophages were massively recruited in *Rnaset2-*expressing C51-derived tumors, with concomitant inhibition of MDSCs and CD206^+^ M2 macrophages recruitment. At later times, a relevant expansion of intra-tumor CD8^+^ T cells was also observed. After re-challenge with C51 parental cells, most mice previously injected with *Rnaset2*-expressing C51 cells still rejected C51 tumor cells, suggesting a Rnaset2-mediated T cell adaptive immune memory response. These results point at T2 RNases as evolutionary conserved oncosuppressors endowed with the ability to inhibit cancer growth in vivo through rebalance of intra-tumor M1/M2 macrophage ratio and concomitant recruitment of adaptive anti-tumor CD8^+^ T cells.

## 1. Introduction

The human *RNASET2* gene encodes for an evolutionarily conserved, pleiotropic extracellular ribonuclease, whose secretion by cancer cells in the tumor microenvironment (TME) is likely involved in tumor suppression. Indeed, two independent in vivo xenograft-based murine models of human ovarian cancer have been previously used to demonstrate a marked *RNASET2*-mediated in vivo tumor suppression [1,2]. It has been previously shown that *RNASET2* affects several important cancer-related parameters (such as modulation of cell proliferation, cytoskeletal re-organization, cell adhesion, motility, and angiogenesis) in a cell-autonomous manner, [3,4,5,6]. Moreover, further studies in both in vitro and in vivo experimental models have demonstrated a marked modulation of macrophage plasticity played by the RNASET2 protein, thus suggesting the occurrence of a *RNASET2*-mediated noncell-autonomous oncosuppressive role [7]. In fact, human *RNASET2* overexpression in two independent human ovarian cancer cell lines, followed by challenging of immunodeficient mouse models with these cells, showed a marked tumor suppressive effect coupled with a *RNASET2*-mediated recruitment of M1-polarized host macrophages within the tumor mass [1,2]. These data, coupled to the well-established notion of T2 RNases as stress response genes [8,9], strongly suggest that human RNASET2 secretion by cancer cells might represent a “danger” signal for cells belonging to the monocyte/macrophage lineage, whose role would be to trigger an effective host immune response. Of note, the macrophage-activating role of T2 RNases has been recently confirmed in species far away from mammals in term of evolutionary distance, such as the invertebrate *Hirudo verbana* [10], thus pointing at T2 RNases as highly conserved immune system-related stress response proteins acting across many Phyla.

The concept that the immune system can recognize and control tumor growth was mainly based on the many in vivo data related to immunoediting phenomenon in preclinical models and humans [11,12], as well as in in vitro studies [13]. This phenomenon mainly points at the importance of CD8^+^ T cells in cancer immunoediting and tumors evading via an adaptive immune resistance phenotype [14]. In the last few decades, this idea has been fully exploited and strengthened, as shown by recent advancement of immunotherapy. The development of immune checkpoint therapy, using blocking antibodies to cytotoxic T lymphocyte antigen-4 (CTLA-4), programmed death-1 (PD-1), or programmed death-ligand 1 (PD-L1), and by chimeric antigen receptor (CAR) T cells has represented a fundamental to stimulate and induce immune effector cells against the tumor that ultimately lead to the elimination of cancer cells [15,16,17,18,19,20].

Macrophages represent key innate immune effectors fighting against pathogens and tumors, but they also have a role in the regulation of tissue homeostasis, repair and tumor progression [21]. These cells can experience a broad spectrum of polarization states in vivo, with different alternative phenotypes in which anti-tumor or pro-tumor activities are represented by M1-like and M2-like cells, respectively [22]. Indeed, based on numerous in vitro experimental results, these cells have previously been too simplistically termed M1- and M2-macrophages [23,24,25], but now, due to a countless new in vivo data obtained from diverse chronic inflammatory diseases including cancer, [22] such dual classification scheme was replaced by a model that envisages a “continuum” of macrophage polarization states characterized by a much broader and heterogeneous transcriptional and functional repertoire [26]. In this new vision pointing out the extreme plasticity of macrophages, when detected in cancer tissues these cells have been defined tumor-associated macrophages (TAMs) with M2-like features. TAMs appear the most abundant tumor-infiltrating inflammatory cells and act as crucial drivers of tumor-promoting inflammation, tumor progression and metastasis [27,28]. Moreover, it has been described that TAMs subtypes can derive from differentiation of monocytic myeloid-derived suppressor cells (MDSCs), adding more complexity to this inflammatory-tumor link [29].

In light of the previously mentioned role of human RNASET2 in modulating the macrophage activation/polarization state, to further investigate the oncosuppressive role of T2 Ribonucleases in the context of a completely immunocompetent experimental model we report here the in vivo role of the murine *Rnaset2* gene by overexpressing it in either mammary adenocarcinoma-derived TS/A or C51 colon carcinoma murine cells and injecting them subcutaneously in syngeneic BALB/c mice, using both empty vector-transfected (E) and parental (P) cells as a control. 

In this work, we report for the first time in a syngeneic mouse model a significant inhibition of tumor growth in mice injected with murine *Rnaset2*-expressing C51 cells in comparison to control tumor cells-injected mice. Furthermore, four out of ten mice were able to completely reject primary full length *Rnaset2*-expressing C51 cells. Intriguingly, three out of four of these tumor-suppressed mice remained tumor-free when re-challenged with parental untransfected C51 cells, indicating the establishment in these mice of immune memory responses and thus, induction of adaptive anti-tumor T immune cells as well. Furthermore, a trend for an in vivo *Rnaset2-*mediated tumor suppressive effect was observed in TS/A cells as well.

To ascertain the role of immune cells in *Rnaset2-*mediated tumor suppression, we thoroughly investigated the tumor cell immune infiltrates in the C51 tumor model. Compared to control mice, M1 macrophages were massively recruited in the *Rnaset2-*expressing tumors at 2 weeks post-tumor inoculation, with a concomitant inhibition of both M2 macrophages and MDSCs influx, whereas at 3 weeks post-injection a statistically significant expansion of intra-tumor CD8^+^ T cells population in mice injected with *Rnaset2-*overexpressing C51 cells was observed. 

Moreover, in C51 tumor-rejecting mice a statistically significant expansion of spleen IFNγ-secreting CD4^+^ T helper (TH) type 1 cells and F4/80^+^ macrophages were observed when compared to both P and E C51-injected control mice. Moreover, a trend for expansion of spleen myeloid CD11b^+^ TNFα -producing cells was also detected.

Taken together, these results support the role of T2 Ribonucleases as evolutionary conserved tumor suppressor genes endowed with the ability to inhibit cancer growth in vivo also in immunocompetent experimental models, through rebalance of intra-tumor M1/M2 macrophage polarization towards TNFα-producing M1-type macrophages and recruitment of adaptive CD8^+^ T cells.

## 2. Results 

### 2.1. Establishment of Mouse Rnaset2-Overexpressing TS/A and C51 Cell Clones and Assessment of Their Proliferation Behavior In Vitro and In Vivo

Murine parental TS/A breast adenocarcinoma (TS/A P) and C51 colon carcinoma (C51 P) cells were first investigated by western blot analysis to assess their endogenous expression level of *Rnaset2*. As shown in Appendix A, both lines showed very low Rnaset2 expression at the protein level, thus making them suitable for gene overexpression-based analysis. Both cell lines were therefore stably transfected with a pcDNA empty vector (TS/A or C51 “E” clones) or a pCDNA vector engineered for murine full-length *Rnaset2* cDNA overexpression (TS/A or C51 “FL *Rnaset2*” clones). After a considerable screening, a few *Rnaset2*-overexpressing clones were obtained for both cell lines (Figure 1A,B) and some were selected for further in vitro and in vivo analysis. 

As depicted in Figure 2A, mouse *Rnaset2*-overexpressing TS/A clones did not show a significant difference in their in vitro proliferation rate when compared to control, empty vector-transfected clones. However, when the same clones were challenged in vivo by subcutaneous injection in syngeneic BALB/c mice, a trend for *Rnaset2*-mediated decrease of tumor growth rate was observed (Figure 2B) even though there was no impact on rate of survival in mice (Figure 2C).

These data are in keeping with our previous results obtained in a xenograft-based human ovarian cancer experimental model, where the highly aggressive Hey3Met2 cell line was engineered to overexpress human *RNASET2* [1]. Indeed, *RNASET2*-mediated tumor suppression was detected in vivo but not in vitro in this cell model.

However, when the same xenograft-based assays were carried out in the less aggressive OVCAR3 cell line, the oncosuppressive role of human *RNASET2* was clearly observed in both in vitro and in vivo experimental settings [2] suggesting that the amplitude of *RNASET2*-mediated tumor suppressive effects was somehow cell line-dependent, possibly based on the intrinsic tumorigenic potential of the chosen cellular model.

Taken together, these data prompted us to further investigate the oncosuppressive potential of murine *Rnaset2* using the less aggressive colon cancer-derived C51 cell line.

Mouse *Rnaset2*-overexpressing C51 clones showed a significantly decreased proliferation rate in vitro when compared to control, empty vector-transfected clones (Figure 3A), unlike TS/A cells. Importantly, there was also a statistically significant retardation of C51 tumor growth in C51 FL *Rnaset2*-injected mice in comparison to both empty vector C51 E and C51 P control mice (Figure 3B), and a significant rate of tumor-free survival mouse (33%, i.e., 2 out of 6 mice) (Figure 3C) that was confirmed with similar results in another set of five injected mice per group (40% of tumor-free survival mouse, i.e., 2 out of 5 mice). Taken together, these findings point at a *Rnaset2*-mediated tumor suppressive effect in the C51 experimental model, which is apparently carried out both in vitro and in vivo. Since these results are in keeping with our previous data obtained with human RNASET2, they clearly suggest that tumor suppression by mammalian members of the T2 RNase family seems to represent an evolutionary conserved biological host defense mechanism. Moreover, unlike our previous investigations on human RNASET2, the syngeneic C51-based model offered us the unprecedented opportunity to further explore the role of T2 RNases in the context of a completely immunocompetent experimental model.

### 2.2. Murine FL Rnaset2 Overexpression in C51 Cells Correlates with Partial Tumor Rejection, with Tumor Microenvironment’s Immune Cells Modulation and Tumor Vessel Inhibition

Next, C51 mouse colon carcinoma cells overexpressing the murine *Rnaset2* gene (C51 FL *Rnaset2*) were inoculated in parallel with parental, untransfected C51 P and empty vector (C51 E) cancer cells in distinct groups of BALB/c mice in order to investigate after two months its in vivo capacity to resist a homologous re-challenge with C51 P cells and the features of infiltrating immune cells.

In the C51 tumor model, four out of ten mice injected with FL *Rnaset2* C51 cells were able to completely reject tumor cells and remained tumor-free for about two months (Figure 4A). In order to determine if an immune response had been established in these tumor-rejecting mice, we proceeded to re-challenge them with the parental untransfected C51 P cells in parallel with control groups of mice, i.e., naïve mice and C51 E injected mice. Strikingly, 75% (three out of four) of mice were now able to reject also the parental C51 P tumor cells (Figure 4B), indicating that an immune memory response was induced in vivo by C51 FL *Rnaset2* engineered cells. Moreover, interestingly, as depicted in Figure 4C, starting on day 14 and afterwards on day 17 and day 20 post-tumor injection, a significant decrease in tumor growth rate in C51 FL *Rnaset2*-injected mice was observed when compared to C51 P-injected control mice. The same trend emerged by comparing C51 E and C51 FL *Rnaset2* recipient mice.

In these differently C51-injected mice we performed an immunohistochemical analysis at day 14 and 20 post-tumor injection, in order to investigate putative immune myeloid, lymphoid, and endothelial cell composition changes in the tumor microenvironment. Strikingly, a marked *Rnaset2*-mediated rebalance of intra-tumor M1/M2 macrophages was detected (Figure 5A). In fact, an increment of CD86^+^ M1 macrophages in C51 FL *Rnaset2*-injected mice was observed in comparison to both C51 E and C51 P control mice at day 14 post-tumor injection, coupled to a more pronounced decrease of CD206^+^ M2 macrophages that remained unchanged until day 20. 

Regarding granulocyte neutrophil counts, we detected no particular modulation of this cell subset in the three groups of mice whereas, more interestingly, MDSCs cell number turned out to be profoundly inhibited in the C51 FL *Rnaset2* group (Figure 5B), suggesting an in vivo role of *Rnaset2* in the modulation of tumor recruitment of MDSCs as well. By staining tumor sections for the CD31 marker, related to the tumor vasculature and to the angiogenesis-associated process, we found a trend for a decrease of vessel numbers in C51 FL *Rnaset2* injected mice, suggesting a role for *Rnaset2* in the inhibition of tumor angiogenesis (Figure 5C). 

Finally, CD4^+^ and CD8^+^ T cells assessment at day 20 post-tumor injection showed a five-fold increase of the CD8^+^ T immune cells in the C51 FL *Rnaset2* group, as shown in Figure 5D, indicating a potential *Rnaset2*-mediated anti-tumor role of these cells as effector cytolytic immune cells. To strengthen the data suggesting a rebalance of M1/M2 macrophages within the tumor, we performed another assay on day 17 after tumor injection on tumor-infiltrating leukocytes, by carrying out flow cytometry assays on digested tumors to assess the proportion of CD86^+^ M1 and CD206^+^ M2 macrophage subsets. As shown in Figure 6 we observed a three-fold increase of the M1/M2 ratio in mice injected with FL *Rnaset2*-overexpressing C51 cells in comparison to C51 P mice and a much more enhanced ratio in comparison to C51 E control mice, thus reinforcing the notion that *Rnaset2* is able to influence the host macrophage compartment towards an increase of the intra-tumor M1/M2 macrophage ratio.

### 2.3. Systemic Immune Cell Investigation Revealed that Rnaset2-Associated Rejection Correlates with Expansion of M1 Macrophages and Triggering of CD4^+^ TH1 Cells

Next, we wanted to investigate if C51 P cells rejection observed in mice previously challenged with C51 FL *Rnaset2* cells was associated with induction of both systemic M1 macrophage and TH1 cell induction. To do this, we assessed percentages of F4/80^+^ macrophages and CD4^+^ and CD8^+^ T cells in spleens of C51 P-rejecting mice and mice receiving either E or P control C51 cells by flow cytometric analysis. Interestingly, as shown in Figure 7A,C, we found that in tumor-rejecting C51 FL *Rnaset2*-injected mice there was a gain of F4/80^+^ mature macrophages as well as a two-fold increase of percentage of CD4^+^ T helper cells, whereas CD8^+^ T cells were not modulated in the different groups investigated.

To further investigate the type of induction of spleen immune response, we analyzed TNFα and IFNγ cytokine production from differentially stimulated spleen cells, i.e., LPS for macrophages and PMA plus ionomycin for T cells. Mice injected with FL *Rnaset2*-overexpressing C51 cells showed an increase of TNFα-producing myeloid CD11b^+^ macrophages in comparison to the other groups of animals, as well as an increment of IFNγ-secreting TH cells in this same group of mice (Figure 7B,D). These findings corroborate the notion that C51 P tumor-rejecting mice following previous C51 FL *Rnaset2* injection undergo a marked induction of systemic M1 macrophages and TH1 type immune cells, both of which are likely important for the whole immune response during rejection and in particular for induction of CD8^+^ T effector lymphocytes.

## 3. Discussion

In the last decades, cancer research has been increasingly focused on the potential role of the TME for both the progression and metastatic spread of most cancer types [28,30,31]. In this context, cells from both the innate and adaptive immune system have long been acknowledged to play a key role in TME-mediated control of cancer growth [27,32,33].

For instance, TAMs represent an abundant TME cell population which is crucially involved in cancer progression by establishing a complex molecular and cellular crosstalk between the immune system, other TME cellular components (such as fibroblasts and endothelial cells) and cancer cells themselves [27,28].

Given their widely acknowledged functional plasticity, macrophages within the TME can carry out either pro-tumor or anti-tumor roles, as epitomized by the well-established M1 vs M2 polarization paradigm [21,22,23,24,25]. However, although a significant bulk of experimental evidence points at TAMs as M2-polarized macrophages, the mechanisms regulating the M1/M2 polarization imbalance in the TME still remains poorly understood.

Recently, we have defined the human *RNASET2* gene as a potential player for in vivo M1/M2 imbalance regulation. Indeed, two independent xenograft-based experimental models of human ovarian carcinoma clearly showed the occurrence of a marked *RNASET2*-mediated in vivo tumor suppressive role, which was apparently based on the ability of cancer cells-derived RNASET2 protein to recruit M1 macrophages within the TME [1,2]. These data are in keeping with a growing number of reports suggesting a role for several members of the T2 ribonuclease family as stress response genes involved in host defense [7]. Moreover, in vitro polarization assays in THP-1 human promyelocityc cell line-derived macrophages have recently confirmed a direct role for human RNASET2 protein in shifting the balance between M1 and M2 macrophages in favor of the former cell type [34]. Therefore, RNASET2 likely represents a novel member of the alarmin family, a rather heterogeneous group of signaling molecules endowed with the ability to sense the occurrence of “dangerous events” at the tissue level (like an ongoing infection or an incipient neoplastic growth) and in turn trigger an adequate host defensive response [7].

Despite being a key player in the control of cancer development and progression, the innate immune system has to establish a tight crosstalk with the adaptive immune system in order to carry out a fully effective role in either directions (i.e., either pro-tumor or anti-tumor). It is therefore tempting to speculate that signaling molecules originally defined by their effects on the innate immune system (such as RNASET2) might actually be involved in triggering an adaptive immune response as well.

To address this topic, we report here for the first time the results from in vivo experiments to decipher the role of the mouse *Rnaset2* gene in syngeneic murine models. To this aim, two different mouse cell lines (breast adenocarcinoma-derived TS/A and colon carcinoma-derived C51) were chosen. Since our previous results indicated that human *RNASET2* can carry out its oncosuppressive role to a different extent depending on the adopted experimental model [1,2] we decided to assess the role of murine *Rnaset2* by means of both in vitro and in vivo assays. 

The in vitro cell proliferation rate and in vivo tumor growth rate in syngeneic BALB/c mice were thus evaluated for FL *Rnaset2*, E vector, and P untransfected cancer cells.

Taken together, the experimental data allowed us to conclude that, whereas the highly aggressive TS/A cell line was only partially sensitive to *Rnaset2*-mediated tumor suppression (which was observed by in vivo assays only), the less tumorigenic C51 cell line turned out to undergo a marked *Rnaset2*-mediated tumor suppression in both in vitro and in vivo experimental settings, and was therefore chosen for further investigation. The observed cell line-dependent effects of murine *Rnaset2* is totally in keeping with our previous results with xenograft-based in vitro and in vivo assays using human RNASET2 [1,2].

Moreover, since our previous investigations on human RNASET2 protein unveiled its ability to modulate in vivo the recruitment of myeloid cells in the TME (in particular by triggering M1-polarized macrophages recruitment in in vivo xenograft-based murine models of human cancer) [1,2], the development of our syngeneic mouse model allowed us to further investigate the relationship of macrophages and other immune cells involved in the anti-tumor immune response. 

In the C51 tumor model, a clear trend for a marked inhibition of tumor growth in mouse FL *Rnaset2*-overexpressing clones compared to control clones was observed, with several time points showing significant difference between FL *Rnaset2* and control C51 E and P cells. C51 P and C51 E control clones also showed a slight difference in in vivo tumor growth rates as well, although statistically not significant. Such difference might be related to either the different culture conditions in which these cells were grown in vitro before being injected in syngeneic mice (i.e., with or without the G418 selective agent for expression vector maintenance) or, alternatively, to empty vector-induced genome integration effects.

It is moreover worth noting that the observed slight difference in the growth rate in vivo between C51 P and C51 E cells, showing a trend for a slightly more pronounced tumor growth in the former, was mirrored in most flow cytometry data shown in Figure 7, where C51 P cells showed a more marked effect than C51 E cells with respect to *Rnaset2*-expressing cells in terms of several immunological-related parameters, which in some cases reached statistical significance. We thus speculate that the significant differences between C51 P and C51 E cells emerged in most single immunological assays carried out by cytometry are in keeping with the non-significant trend emerged in in vivo for the much complex process of growth rate assay. 

Thus, taken together we reckon that, notwithstanding a few discrepancies (such as the higher M1/M2 ratio in C51 P vs. C51 E tumors shown in Figure 6) our data provide a strong evidence about the role of the murine immune system in *Rnaset2-*mediated tumor suppression, also in light of the wide amount of literature data supporting such role for several members of the T2 RNase family.

For this reason, we decided to further investigate the mechanisms involved in tumor growth delay in the C51 tumor model. Interestingly, four out of ten C51 FL *Rnaset2* mice were able to completely suppress tumor cells growth and remained tumor-free for about two months. When these four mice were then re-challenged with parental C51 P cells, 75% of them became tumor protected, whereas control mice were not, indicating immune cell adaptive induction of memory T cell responses.

We next analyzed, at two different time post-inoculation, immune cell infiltration of tumor masses of mice injected differently with C51 FL *Rnaset2*, C51 E and C51 P cells. Immunohistochemistry analysis showed a clear rebalance (i.e., an increment) of M1/M2 polarization ratio in C51 FL *Rnaset2*-injected mice, concomitant with inhibition of MDSCs, granulocytes and vessel numbers. The myeloid cell compartment appears to be a crucial determinant in the fate of tumor growth [27], indeed M2-like macrophages and/or TAMs as well as MDSCs associate with worse tumor clinical outcome, tumor dissemination, angiogenesis, immunosuppression, and metastasis [21,35,36]. Concerning tumor angiogenesis, it has been reported that several derivative peptides from human RNASET2 protein structure can exert antiangiogenic effects using HUVEC tube formation assay and using ex vivo CAM assay, and this effect is dependent on the peptide’s ability to bind actin [37]. Moreover, due to its intracellular and extracellular multifunctional roles, and in particular its antiangiogenic and antitumorigenic capacities, the RNASET2 protein represents a high therapeutic potential candidate for anticancer therapies [38].

Conversely, M1-like macrophage induction correlates with better control of tumor growth and also stimulation of anti-tumor protective adaptive T cells [39,40], in particular with CD4^+^ TH1 cells, which are key activators in the priming phase, and CD8^+^ T cytolytic cells in the rejection effector response against tumor [41,42,43,44].

To corroborate our results showing an in vivo *Rnaset2*-mediated rebalance of M1/M2 macrophages ratio, we conducted a flow cytometric assay on day 17 after tumor injection on tumor-infiltrating leukocytes in the three groups of C51 tumor model. We demonstrated that there was an increase of CD86^+^ M1/CD206^+^ M2 macrophage ratio in the TME of C51 FL *Rnaset2* injected mice when compared to C51 E and P control tumors. This point is fundamental since reprogramming TAMs towards an M1 anti-tumor role actually represents one of most considered strategies for immune control cancer growth [27,32,39,40].

IFNγ has a crucial role in the orchestration of an effective anti-tumor response, mainly by a direct role by decreasing proliferation and metabolic activity in tumor cells, as well as indirectly by up-regulating MHC class I and II molecules expression, inhibiting angiogenesis, and driving TH1 polarization by DCs [45,46,47].

We therefore proceeded to characterize functional systemic immune responses in tumor-rejecting mice, in comparison to control tumor-bearing mice, studying in vitro activation of macrophages and T cells from spleens of differently groups of injected mice. Noteworthy, we found that *Rnaset2*-associated memory rejection correlates with systemic expansion of F4/80^+^ TNFα-producing M1 macrophages and triggering of CD4^+^ TH1 cells both in order of increment of their percentage and also for IFNγ-secreting cells. 

Taken together, these results indicate that overexpression of murine *Rnaset2* in a colon syngeneic mouse carcinoma model leads to a concerted anti-tumor immune response able to exert a tumor growth delay and rejection. Of note, this immune response appeared correlated with induction of innate cell compartment in the TME, in particular modification of M1/M2 balance ratio, decrease of number of MDSCs, tumor vessels inhibition and triggering adaptive immune CD8^+^ T effector cells.

## 4. Materials and Methods

### 4.1. Cloning of the Murine Rnaset2 cDNA in pcDNA3 Plasmid 

The pcDNA3 plasmid (Invitrogen) was chosen as backbone expression vector. The inducible vector was purchased from the VectorBuilder company (Appendix A) and used as a template for a PCR reaction with the following primer pair for inserting the 3×FLAG sequence into the pcDNA3 vector:
3xFLAG_version2 Rev: 5′ AGAGGTTCTAGACTACTTGTCATCGTCATCCTTGTAATC 3′3xFLAG _version2 Fw: 5′ AACTGGCTCGAGGACTACAAAGACCATGACGGTGA 3′

The amplification product was digested with XhoI and XbaI, gel purified and cloned into the pcDNA3 vector before transforming in the DH5α *E. coli* strain. Plasmid DNA was purified and then sent to sequencing (BMR, Padova, Italy), before using it for the following steps. Different constructs, already available in our lab, bearing the murine *Rnaset2* gene were used as templates for PCR reactions with the following primers:
mRnaset2A Fw: 5′ TGCTAAGGATCCACCATGGCGCCG 3′mRnaset2A Rev: 5′ ACCTGAGAATTCATGTTGGGTCTTTGTAGGTGGA 3′mRnaset2A overlap Fw: 5′ AGCCGGGGGAGCAGCTGTCCTCCAGGCAGGAA 3′mRnaset2A overlap Rev: 5′ CTGCTTGGGGGACGGCTGCTCCCCTGGCTCA 3′

The amplification products were digested with EcoRI and BamHI, gel purified and cloned into the pcDNA3 vector (together with a synthetic oligo coding for a TEV protease recognition site with EcoRI and XhoI sticky ends) before transforming in the DH5α *E. coli* strain. 

Collectively, the inserted coding region is represented by the full length murine *Rnaset2* gene (FL *Rnaset2*) followed by TEV and 3xFlag sequences. Plasmid DNA was purified and then sent to sequencing (BMR, Padova, Italy), before using it for the transfections. The resulting plasmid, together with its empty control counterpart (E), was used to stably transfect two BALB/c-derived murine cancer cell lines, TS/A (mammary adenocarcinoma) and C51 (colon origin).

### 4.2. Animal Tumor Models

TS/A mammary adenocarcinoma and C51 mouse colon adenocarcinoma (Parental, P) cells (European Collection of Cell Cultures, Sigma-Aldrich), both of them of BALB/c origin, were cultured in DMEM with 10% FBS at 37°C, 5% CO_2_ atmosphere. 

Stably-transfected clones were cultured in the basic medium supplemented with G418 (300 μg/mL for TS/A and 400 μg/mL for C51). TS/A and C51 (2 × 10^5^) cells were subcutaneously (s.c.) implanted in 8- to 10-week-old immunocompetent syngeneic BALB/c mice (Envigo, Cambridgeshire, UK). Besides empty pcDNA3 vector-transfected cells, parental untransfected cells (labelled C51 P and TS/A P, respectively) have also been added to the experimental plan in in vivo cell transplantation assays, in order to include a further negative control assay. In particular, we wanted to exclude unpredictable in vivo side-effects due to random integration in the transfected cell’s genome of the empty vector. The tumor volume was determined using the formula: (d) 2 × D × 0.52, where d and D, are the short and long dimension (cm) of the tumor, respectively, measured with a caliper. Mice were bred and kept under standard conditions at the animal facility of the IRCCS Ospedale Policlinico San Martino. Housing, treatment, and killing of animals followed national legislative provisions for the protection of animals used for scientific purposes and an ethics committee approved the protocol (#299/2018-PR). Euthanasia was performed when the tumor reached a volume of about 1.5 cm^3^ by cervical dislocation after CO_2_ treatment.

### 4.3. Cell Proliferation Assay

The Cell Titer^®^ 96 Non-Radioactive kit (Promega, Madison, WI, USA) was used to determine viable cell number based on the cellular conversion of a tetrazolium salt into a formazan product. The absorbance of the solubilized formazan product (directly proportional to the number of cells) was recorded using a 96-well plate reader daily over a 7- or 10-day period. Samples were seeded in triplicates.

### 4.4. Western Blot Analysis 

Adherent cells were mechanically scraped in PBS in the presence of 5 mM EDTA and resuspended in lysis buffer (0.5% Igepal, 0.5% Triton X-100 and 5 mM EDTA in PBS) supplemented with protease inhibitors cocktail (PMSF, benzamidine, aprotinin, and leupeptin). Quantification of total proteins was performed with Bradford reagent (BIO-RAD, Kidlington, UK), using bovine serum albumin as standard. For the SDS-PAGE analysis, 70 μg of intracellular lysate were loaded. Immunoblot analysis was performed using standard procedures and detected with a chemiluminescent substrate, WESTAR ETA C ULTRA 2.0, Cyanagen (Sigma Aldrich, St. Louis, MO, USA). Antibodies used were: Rabbit monoclonal anti-FLAG (Sigma Aldrich); HRP anti-rabbit IgG (Jackson ImmunoResearch Laboratories, West Grove, PA, USA).

### 4.5. Lymphoid and Myeloid Organ Isolation

Spleen cells were isolated by scissor dissociation, collected in RPMI 1640–10% FBS, and filtered using a 70 μm cell strainer (Becton Dickinson (BD), San Diego, CA, USA). Erythrocytes in the splenic preparations were removed by hypotonic lysis with NH4Cl–KHCO3–EDTA solution. Tumor infiltrating leucocytes from C51 colon carcinoma were obtained by cutting the tumor mass into small pieces followed by incubation (1 g/10 mL) with a mixture of enzymes dissolved in serum free RPMI: collagenase type I (0.5 mg/mL), collagenase type IV (0.5 mg/mL), and hyaluronidase (0.25 mg/mL), all from Sigma Aldrich, and DNase I (0.1 mg/mL) from Roche (Mannheim, Germany) for 30 min at 37 °C. Undigested material was removed by filtration using a 70 μm cell strainer (BD) and the single cell suspensions were recovered and washed by centrifugation in RPMI 1640–10% FBS to remove residual enzymes.

### 4.6. Flow Cytometry Analysis

Spleen cells or tumor infiltrating leucocytes were first incubated with an FcR blocking reagent (Miltenyi Biotec, Bergisch Gladbach, Germany) for 10 min. Afterward, cells were labeled with anti-CD4 (clone GK1.5), anti-CD8 (clone 53-6.7), and anti-CD11b (clone M1/70) mAbs, all from eBioscience (San Diego, CA, USA); anti-F4/80 (clone BM8) and anti-CD86 (clone GL-1) mAbs from BioLegend (San Diego, CA, USA), and anti-CD206 (clone MR5D3) mAb from BIO-RAD. Briefly, after physical parameter setting (Forward and Side Scatter), and using anti-LCA CD45 (clone 30-F11) mAb from BioLegend mAb for tumor infiltrating leucocytes analysis, lymphocyte and myeloid populations were identified and then M1, M2, macrophages and T cells were evaluated by mAb-specific staining and assessed by FACS analysis using CELLQUEST or DIVA software (BD). The direct immunofluorescence was performed using FITC-, PE-, or allophycocyanin-conjugated specific mAbs. Negative controls included directly labeled FITC-, PE-, and APC-conjugated isotype-matched irrelevant mAbs (BD).

### 4.7. Intracellular Staining Analysis

Spleen cells from different groups of BALB/c mice were analyzed for cytokine content as previously described [48]. Briefly, T cells were seeded at 10^6^/mL in RPMI 1640–10% FBS, and stimulated with 10 ng/mL phorbol 12-myristate 13-acetate (PMA, Sigma-Aldrich) and 500 ng/mL ionomycin (Sigma-Aldrich) in the presence of monensin (BD) during 4-h incubation at 37 °C, and then treated with Cytofix/Cytoperm fixation and permeabilization kit (BD). For macrophage stimulation, spleen cells were seeded at 10^6^/mL in RPMI 1640–10% FBS, and stimulated with LPS 100 ng/mL (Sigma-Aldrich) in the presence of monensin during 4 h incubation at 37 °C. The permeabilized cells were intracellularly stained with the PE-conjugated cytokine-specific mAbs anti-IFNγ (clone XMG1.2), and anti-TNFα (clone MP6-XT22), all from BD; or isotype-matched irrelevant mAbs from BioLegend.

### 4.8. Immunohistochemistry of Tumor Samples

Cryostat sections (6 μm thick) were air-dried and fixed in cold acetone for 10 min. Immunostaining was performed using a previously described procedure [43] and the following primary mAbs: anti-CD4 and anti-CD8 T lymphocytes (DITTA), anti-CD31 (clone MEC 13.3, kindly provided by A. Mantovani, Humanitas Institute, Milan), anti-granulocyte Gr-1 (clone RB6-8C5), anti-myeloid cells CD11b (clone M1/70) from ImmunoKontact (Oxford, UK); anti-M1-type CD86 (clone PO.3) and anti-M2-type CD206 (clone MR5D3), both from AbD Serotec (Oxford, UK). The Alexa Fluor 594 goat anti-rat from ThermoFisher was used for CD11b as secondary antibody. Quantitative studies of stained sections were performed independently by three researchers in a blinded fashion. Cells and vessels counting was carried out in 8–12 randomly chosen fields under a Leica Wetzlar light microscope (Solms, Germany) or an ApoTome microscope with AxioCam (Karl Zeiss, Thornwood, NY, USA) at 400× magnification, 0.180 mm^2^/field. The results are expressed as cells or vessels number per high magnification microscopic field (cells or vessels no./HMMF, mean ± SE). Images of slides were acquired using Aperio AT2 scan (Leica).

### 4.9. Statistical Analysis

Data were analyzed for statistical significance using two-way ANOVA with Bonferroni’s posttest or nonparametric Mann–Whitney by Prism Graphpad software (GraphPad Software Inc., La Jolla, CA, USA). All error bars represent SEM. Values of **** *p* < 0.0001, *** *p* < 0.001, ** *p* < 0.01, and * *p* < 0.05 were considered significant. 

## 5. Conclusions

In this work, we provide the first in vivo evidence supporting the role of a member of the highly conserved T2 ribonuclease gene family in tumor suppression by means of the functional involvement of molecular and/or cellular effectors belonging to both the innate and adaptive immune systems. 

By developing a murine experimental model where C51 mouse colon carcinoma-derived cells were engineered to overexpress the mouse *Rnaset2* gene and subsequently challenged in syngeneic BALB/c mice, we could confirm the marked tumor suppressive role of this gene (as already reported for its human *RNASET2* ortholog), but we also unveiled that *Rnaset2*-mediated tumor suppression is associated not only with an increase in the intratumor M1/M2 macrophages ratio, but also with the establishment of a tumor suppressive immunological memory, which clearly entails adaptive immunity as a further biological weapon exploited by this highly conserved, pleiotropic gene to carry out its oncosuppressor activity.

## Figures and Tables

**Figure 1 cancers-12-00717-f001:**
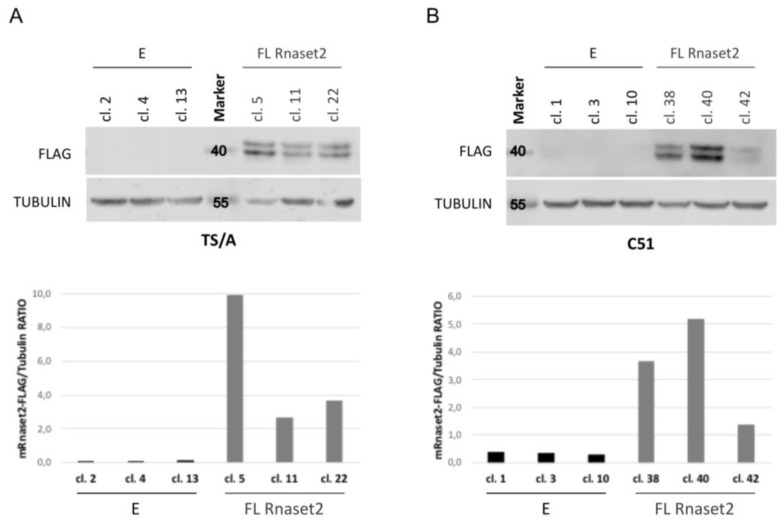
Immunoblot analysis of murine *Rnaset2*-overexpressing clones in TS/A (panel **A**) and C51 (panel **B**) tumor cell lines. The images showing the whole-filter picture from the western blots shown above are provided in Appendix A.

**Figure 2 cancers-12-00717-f002:**
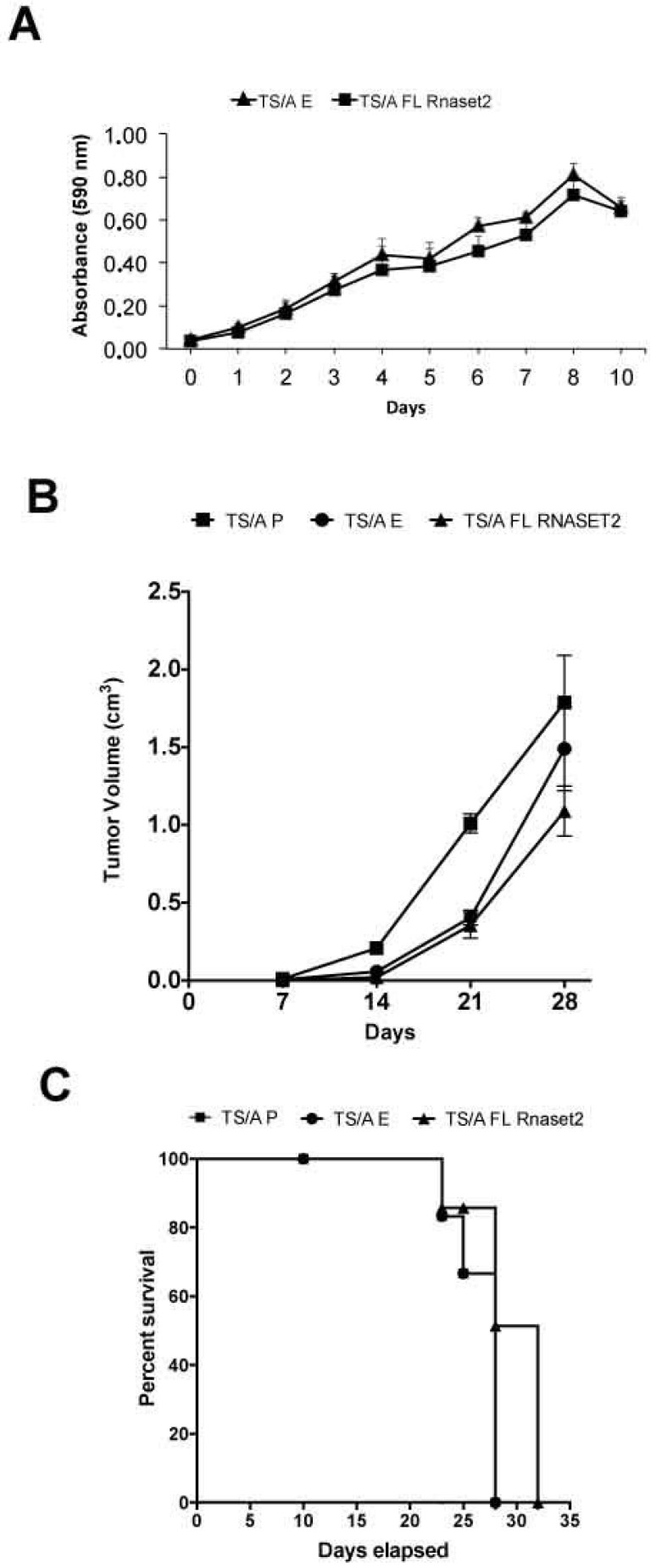
Assessment of in vitro cell proliferation rate (MTT assay) in full length *Rnaset2*-overexpressing TS/A clones (FL *Rnaset2*) in comparison to empty vector-transfected TS/A clones (TS/A E) (panel **A**) and in vivo analysis of tumor growth in BALB/c mice groups (*N* = six animals in group) injected with TS/A P, TS/A E, and TS/A FL *Rnaset2* cell lines (panel **B**). Survival curves versus time (days) of BALB/c mice groups injected with TS/A P, TS/A E, and TS/A FL *Rnaset2* cell lines (panel **C**).

**Figure 3 cancers-12-00717-f003:**
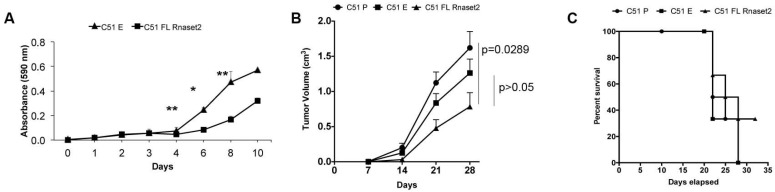
Assessment of in vitro cell proliferation rate (MTT assay) in full length *Rnaset2*-overexpressing C51 clones (FL *Rnaset2*) in comparison to empty vector-transfected C51 clones (C51 E) (panel **A**); in vivo analysis of tumor growth in BALB/c mice groups (*N* = six animals in group) injected with C51 P, C51 E, and C51 FL *Rnaset2* cell lines (panel **B**); survival curves versus time (days) of BALB/c mice groups injected with C51 P, C51 E and C51 FL *Rnaset2* cell lines (panel **C**). For MTT assay and tumor growth, an ANOVA statistical analysis was performed assuming *p* < 0.05 as a threshold value. * *p* < 0.05; ** *p* < 0.01.

**Figure 4 cancers-12-00717-f004:**
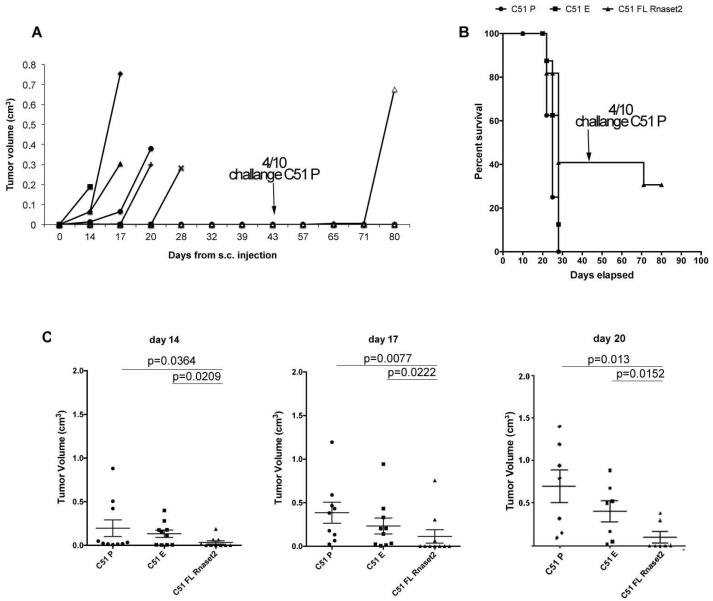
Tumor growth follow up of ten mice injected with C51 FL *Rnaset2* up to about two months and subsequent follow up of tumor growth in four tumor-free of them receiving a C51 P tumor challenge until day 80 from the first C51 FL *Rnaset2* injection (panel **A**); survival curves versus time (days) of BALB/c mice groups injected with C51 P, C51 E, and C51 FL *Rnaset2* cell lines (panel **B**); in vivo analysis of tumor growth with statistical difference at three time points (day 14, 17, and 20) of other groups (*N* = ten animals in group) of BALB/c mice injected with C51 P, C51 E, and C51 FL *Rnaset2* cell lines (panel **C**). Horizontal bars indicate mean ± SE (Standard error) values for each group. *p* values of statistically significant differences between the groups connected by lines are also reported. An ANOVA statistical analysis was performed assuming *p* < 0.05.

**Figure 5 cancers-12-00717-f005:**
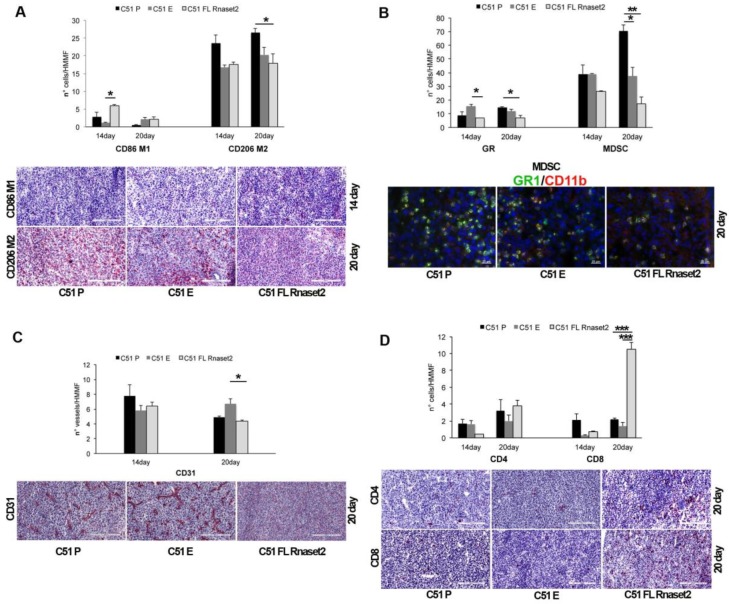
Quantification of different key markers of tumor infiltrating immune cells, as follows: CD86^+^ M1 and CD206^+^ M2 (panel **A**), Gr-1 (GR) for granulocytes and CD11b^+^Gr-1^+^ for MDSCs (panel **B**), CD31 for vessels (panel **C**), and CD4 and CD8 for T cells (panel **D**) per HMMF in BALB/c mice injected with C51 P, C51 E, and C51 FL *Rnaset2* cell lines (*N* = six animals in group). The mean ± SE are indicated. Immunohistochemistry and immunofluorescence analyses of tumor sections from BALB/c mice injected with C51 P, C51 E, and C51 FL *Rnaset2* cell lines at indicated time points are shown. An ANOVA statistical analysis was performed assuming *p* < 0.05 as a threshold value. * *p* < 0.05; ** *p* < 0.01; *** *p* < 0.001.

**Figure 6 cancers-12-00717-f006:**
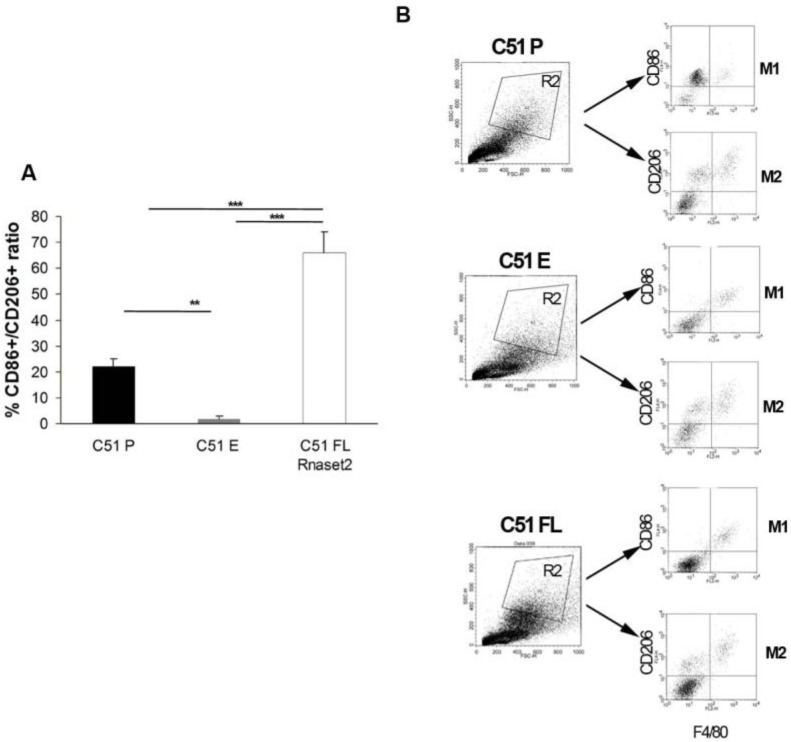
Cytometric flow assessment after in vitro tumor digestion has been performed to study percentage of intra-tumor CD45^+^ CD86^+^ M1 and CD206^+^ M2 macrophage ratio in BALB/c mice (*N* = three animals in group) injected with C51 P, C51 E and C51 FL *Rnaset2* cell lines (panel **A**); gating strategy for flow cytometry (FACS) analysis of surface markers in M1 (F4/80^+^ CD86^+^) and M2 (F4/80^+^ CD206^+^) macrophages (panel **B**). An ANOVA statistical analysis was performed assuming *p* < 0.05 as a threshold value. * *p* < 0.05; ** *p* < 0.01; *** *p* < 0.001.

**Figure 7 cancers-12-00717-f007:**
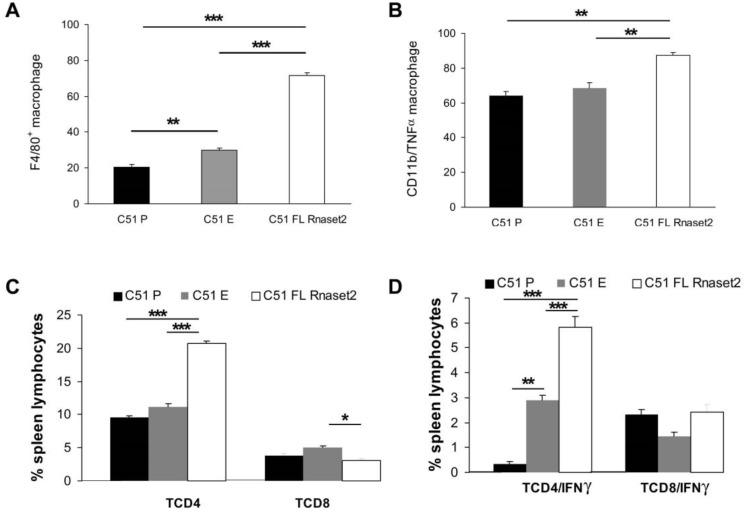
Cell cytometric flow assessment in spleens from groups of BALB/c mice (*N* = three animals in group) at one month after tumor injection, as follows, i.e., receiving C51 P, C51 E tumor cells lines and C51 challenge in rejecting and tumor-free C51 FL *Rnaset2* mice. The cytometric flow analysis has been conducted after in vitro tumor digestion to quantify: Percentage of tumor cell infiltration of F4/80^+^ mature macrophages (panel **A**); percentage of TNFα-secreting CD11b^+^ macrophages by intracellular staining, following in vitro 4 h LPS stimulation (panel **B**); percentage of CD4^+^ and CD8^+^ T cells (panel **C**); and percentage of IFNγ-producing CD4^+^ and CD8^+^ T cells by intracellular staining, following in vitro 4 h PMA plus ionomycin stimulation (panel **D**). An ANOVA statistical analysis was performed assuming *p* < 0.05 as a threshold value. * *p* < 0.05; ** *p* < 0.01; *** *p* < 0.001.

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
