# Peer review of "Overexpression of Murine Rnaset2 in a Colon Syngeneic Mouse Carcinoma Model Leads to Rebalance of Intra-Tumor M1/M2 Macrophage Ratio, Activation of T Cells, Delayed Tumor Growth, and Rejection"

_cancers, 2020, doi:10.3390/cancers12030717_

Round 1

Reviewer 1 Report

Dear Authors, 

The manuscript titled "Overexpression of murine Rnaset2 in a colon syngeneic mouse carcinoma model leads to rebalance of intra-tumor M1/M2 macrophage ratio, activation of T cells, delayed tumor growth and rejection" is a well and logically constructed paper.

In my opinion the introduction could be improved just pointing out the role of immunotherapy in decreasing tumor growth, mentioning papers that have discussed it in vivo (animals, humans) as well as in vitro. 

Furthermore, I suggest you to improve the quality of Figure 2A and Figure 3A to better visualize the differences, maybe reducing the line thickness or changing the colors.

However, I have no major revision to address.

Author Response

REVIEWER N.1

Comments and Suggestions for Authors

Dear Authors, 

The manuscript titled "Overexpression of murine Rnaset2 in a colon syngeneic mouse carcinoma model leads to rebalance of intra-tumor M1/M2 macrophage ratio, activation of T cells, delayed tumor growth and rejection" is a well and logically constructed paper.

In my opinion the introduction could be improved just pointing out the role of immunotherapy in decreasing tumor growth, mentioning papers that have discussed it in vivo (animals, humans) as well as in vitro

Following the reviewer’s suggestion we have modified the Introduction and added this paragraph with new references (already inserted in the test) as follows:

“The concept that the immune system can recognize and control tumor growth was mainly based on the many in vivo data related to immunoediting phenomenon in preclinical models and humans [11,12], as well as in in vitro studies [13]. This phenomenon mainly points at the importance of CD8+ T cells in cancer immunoediting and tumors evading via an adaptive immune resistance phenotype [14]. In the last few decades, this idea has been fully exploited and strengthened, as shown by recent advancement of immunotherapy. The development of immune checkpoint therapy, using blocking antibodies to cytotoxic T lymphocyte antigen-4 (CTLA-4), programmed death-1 (PD-1), or programmed death-ligand 1 (PD-L1), and by chimeric antigen receptor (CAR) T cells has represented a fundamental to stimulate and induce immune effector cells against the tumor that ultimately lead to the elimination of cancer cells [15-20].”

Furthermore, I suggest you to improve the quality of Figure 2A and Figure 3A to better visualize the differences, maybe reducing the line thickness or changing the colors.

Based on the reviewer's suggestion, we have improved the quality of all figures, especially Figure 2A and Figure 3A as depicted in the new version of the manuscript.

However, I have no major revision to address.

Reviewer 2 Report

In this manuscript, the authors showed the tumor suppressive role of Rnaset2 in a murine colon cancer cell line C51-derived subcutaneous mouse model. In addition, they also try to study the potential immunoregulatory role of Rnaset2 in determining colon cancer outcome via regulating macrophages, MDSCs and/or T cells. It is interesting to explore the potential role of Rnaset2 in innate immune regulation as well as memory responses due to the rejection towards the tumor re-challenge. However, the data on the immune cell analysis is not enough and the conclusion is not well supported by the current data. Obvious differences in immune cell changes between the parental cell line and empty-vector expressed cell line are quite confused.

Major points:

  1. Figure 3 C: no p value was shown between C51P and C51 FL Rnaset2 on day 20 which may due to the variation that need to be improved by additional mouse number. Besides, the tumor weight data and ratio of tumor-free mice in each group should be shown.
  2. The data shown in Figure 3 can not support the description on page 6, second last paragraph ‘ … clearly suggested either an in vivo Rnaset2-mdieated’.
  3. Figure 5: the IHC analysis with one marker only can not represent the immune cell types. For example, CD86 can be expressed by monocytes, dendritic cells as well as MDSC to some extent. Multi-color flow cytometry analysis in tumors would be more accurate. Also, the demonstrated photo of IHC staining should be presented.
  4. Figure 5: why the CD206, GR and CD31 stating showed obvious differences between C51P and C51E tumors?
  5. Figure 6: a demonstration figure of gating strategy is needed.
  6. Also, why C51P and C51E displayed significantly difference in figure 6 and figure 7? Since no significant difference was showed in tumor growth between these two groups, is it indicated that the immune cell changes may not be a significant mechanism of Rnaset2-mediated tumor suppression?
  7. The correlations among tumor weight/volume, the expression level of Rnaset2, M1/M2 macrophages, MDSC, Th cell as well as CD8+ T cells should be analyzed to consolidate the conclusion.
  8. It is also interesting to measured the memory cell proportion in the tumor-free mouse. 

Minor points:

Figure 2B, need to label p value.

Author Response

REVIEWER N.2

Comments and Suggestions for Authors

In this manuscript, the authors showed the tumor suppressive role of Rnaset2 in a murine colon cancer cell line C51-derived subcutaneous mouse model. In addition, they also try to study the potential immunoregulatory role of Rnaset2 in determining colon cancer outcome via regulating macrophages, MDSCs and/or T cells. It is interesting to explore the potential role of Rnaset2 in innate immune regulation as well as memory responses due to the rejection towards the tumor re-challenge. However, the data on the immune cell analysis is not enough and the conclusion is not well supported by the current data. Obvious differences in immune cell changes between the parental cell line and empty-vector expressed cell line are quite confused.

Major points:

Figure 3 C: no p value was shown between C51P and C51 FL Rnaset2 on day 20 which may due to the variation that need to be improved by additional mouse number. Besides, the tumor weight data and ratio of tumor-free mice in each group should be shown.

We agree with reviewer 2. Figure 3C has now been improved and included in Fig. 4C. We checked this point, and effectively there was not statistically difference at this time point. Moreover, a single mouse, which had a large tumor over, has been now discarded from the analysis. The reason why it appears there were few mice depends in part from the fact that some mice have been sacrificed to allow IHC analyses for the three times post tumor injection. Albeit new mice per group were studied for IHC at these time points. However, we think that in our assays with several groups of mice used, we detected statistically significance between tumor volumes also between C51P and C51 FL Rnaset2, as shown in figure 4C.

The data shown in Figure 3 can not support the description on page 6, second last paragraph ‘ … clearly suggested either an in vivo Rnaset2-mdieated’.

The phrase in the mentioned paragraph has been modified according to the reviewer’s suggestions as follows:

“Taken together, these findings point at a Rnaset2-mediated tumor suppressive effect in the C51 experimental model, which is apparently carried out both in vitro and in vivo. Since these results are in keeping with our previous data obtained with human RNASET2, they clearly suggest that tumor suppression by mammalian members of the T2 RNase family seems to represent an evolutionary conserved biological host defense mechanism. However, unlike our previous investigations on human RNASET2, the syngeneic C51-based model offered us the unprecedented opportunity to further explore the role of T2 RNases in the context of a completely immunocompetent experimental model.”

Figure 5: the IHC analysis with one marker only can not represent the immune cell types. For example, CD86 can be expressed by monocytes, dendritic cells as well as MDSC to some extent. Multi-color flow cytometry analysis in tumors would be more accurate. Also, the demonstrated photo of IHC staining should be presented.

We agree with the reviewer’s point and now we have inserted all principal image data regarding IHC assays in the revised version of article.

Figure 5: why the CD206, GR and CD31 stating showed obvious differences between C51P and C51E tumors?

This point raised by reviewer is interesting. Indeed, C51P and C51E control clones injected in mice showed a slight difference in some tumor infiltrate immune cells such as CD206+ M2 macrophages, granulocytes and CD31+ cells. Parental, untransfected C51 control cells have been added to the experimental plan in order to include a further negative control assay. In particular, we wanted to exclude unpredictable side-effects due to random integration in the mouse genome of empty pcDNA3 control plasmid in transfected cells. However, the in vivo growth kinetics showed a slight inhibition of C51 E tumor growth, albeit not significant, that may suggests a weak tumor suppressive effect in mice carrying empty vector versus C51P untransfected cells, and in this vision it might be related to either the different culture conditions in which these cells were grown in vitro before being injected in syngeneic mice (i.e., with or without the G418 selective agent for expression vector maintenance) or, alternatively, to the above-mentioned empty vector-induced genome integration effects. However, it should be noted that the slight differences between the two cell types (i.e. M2 and GR) and CD31 endothelial cells between the empty vector cells and C51 P are not statistically significant. But further future investigations will be pursued that could allow to clarify this criticism, even if the interest of it is of moderate importance, in relation to no evident effect on the adaptive immune T cell compartment.

Figure 6: a demonstration figure of gating strategy is needed.

In accordance with the reviewer’s suggestion we put in the revised manuscript the gating strategy employed in the study using flow cytometry assay (see now Figure 5).

Also, why C51P and C51E displayed significantly difference in figure 6 and figure 7? Since no significant difference was showed in tumor growth between these two groups, is it indicated that the immune cell changes may not be a significant mechanism of Rnaset2-mediated tumor suppression?

We thank the referee for this comment. As mentioned above, parental, untransfected C51 control cells have been added to the experimental plan in order to include a further negative control assay. In our opinion, the in vivo growth kinetics strongly suggests a clear tumor suppressive effect in mice carrying Rnaset2-transfected C51 cells versus both C51P untransfected and C51E empty vector-transfected cells.

C51P and C51E control clones actually showed a slight difference in in vivo tumor growth rates as well, although statistically not significant. Such difference might be related to either the different culture conditions in which these cells were grown in vitro before being injected in syngeneic mice (i.e., with or without the G418 selective agent for expression vector maintenance) or, alternatively, to the above-mentioned empty vector-induced genome integration effects.

In any case, the observed slight difference in the growth rate in vivo between C51P and C51E cells, showing a trend for a more pronounced tumor growth in the former, was mirrored in most flow cytometry data shown in figure 7, where C51P cells showed a more marked effect than C51E cells with respect to Rnaset2-expressing cells in terms of several immunological-related parameters, which in some cases reached statistical significance. We thus reckon that the significant differences between C51P and C51E cells emerged in most single immununological assays carried out by flow cytometry and IHC are in keeping with the non-significant trend emerged in in vivo for the much complex process of growth rate assay.

The only exception to this general trend regards the much higher M1/M2 ratio in C51P vs. C51E tumors shown in figure 6. Although we honestly do not have a clear explanation for this observation, we nevertheless can infer from the same figure the even more dramatic effect of Rnaset2-expressing C51 cells on this parameter when compared to both C51P and C51E tumors.

We therefore reckon that, notwithstanding these discrepancies, our data provide quite convincing evidence about the role of the murine immune system in Rnaset2-mediated tumor suppression, also in light of the wide amount of literature data in support of such role for several members of the T2 RNase family.

The correlations among tumor weight/volume, the expression level of Rnaset2, M1/M2 macrophages, MDSC, Th cell as well as CD8+ T cells should be analyzed to consolidate the conclusion.

We agree with reviewer’s criticism, however generally this type of correlation found interest with high number of samples analyzed. However, we tried to put together our data and performed a new statistical assay between tumor weight and CD4, CD8 T cells and MDSC counts, but we did not reach statistical significance.

It is also interesting to measured the memory cell proportion in the tumor-free mouse.

Concerning this point raised by the reviewer, although we agree that it is certainly interesting to study the immunological components of memory more deeply in the tumor-free mice, we believe that in this present work, we have investigated the general and crucial points of Rnaset2 behavior in both in vitro and in vivo in a syngeneic mouse model for the first time.

It will be our future task in a new study to investigate the importance and involvement of cellular subpopulations such as CD4+ and CD8+ T, but also macrophages, granulocytes, NK and DCs in the phenomenon of tumor rejection (by assays employing in vivo depletion of several distinct key immune cells through injection of specific mAbs) and memory capacity (i.e. how long the rejection capacity lasts, i.e. 2-3-6 months?), and the feature of adoptive cell transfer in Winn assays, in order to decipher unambiguously, which immune cell subset is fundamental to transfer protection against tumor challenges. But for all these assays we will commit a large number of new animals and time, in future experiments of a structured project and hopefully this will allow us to add new details in this interesting field of tumor immunology and oncosuppressive role of Rnaset2 protein in relation to the modulation of innate cells such as M1 and M2 macrophages, as well as tumor-specific adaptive T cells.

Minor points:

Figure 2B, need to label p value.

We re-perform evaluation of statistical assay, concerning results in Figure 2, and we did not reach any significance, i.e. p>0.05, and for this reason, we did not add the p value.

Reviewer 3 Report

In this manuscript De Vito et al. analyze the role of Rnaset2 in modulating tumor growth using syngeneic mouse models. The authors first demonstrate that Rnaset2 over expression leads to reduced tumor volume in both a breast adenocarcinoma and a colon carcinoma model. They proceed to show that following rechallenge with parental cells, 75% of mice tested did not develop a new tumor. Finally, they conclude by examining the immunophenotype of the tumor associated macrophages, tumor infiltrating T-cells, and splenic cells.

This work is the first to study the effects of Rnaset2 in an in vivo immunocompetent setting and demonstrates that Rnaset2 expression can lead to an adaptive immune response which can protect from rechallenge, which is a novel finding that will be of interest to the field. However, there are many concerns about the methodology, rigor, and data interpretation that prevent me from recommending this manuscript for publication in its current form. Critically, the authors do not provide data on mouse survival or randomization of their study, and use unacceptably small numbers of mice without including any statistical justification.    

  • Figures do not look publication quality and all need reformatted
  • All figure legends need to include number of independent replicates, statistical tests used, and to indicate what p value markers of statistical significance correspond to.
  • More information about all mouse studies needs to be included. At a minimum: was power analysis preformed, was study randomized, and how was survival impacted.
  • More information is needed about how mice were selected for downstream experiments following tissue harvest.
  • Figure 1: Lacks quantification of western blots.
  • Figure 1: It is unclear which group Cl.1 belongs to
  • Figure 1: Probing for FLAG does not address the question of the amount of Rnaset2 overexpression. Need to probe using antibody that recognizes endogenous protein.  If unavailable, qRT PCR data would also be acceptable.
  • Figure 2: Needs to also include mouse survival data.
  • Figure 2: While not required, images of mouse tumors would help manuscript
  • Figure 2B: Unclear if there is any statistical significance
  • Figure 3: Legend is vague about which groups have 10 vs 6 mice
  • Figure 3b: Is there any statistical significance?
  • Figure 3c: There are clearly outlier mice at the day 17 timepoint, if these mice are removed are the results still statistically significant? Also were these mice used for downstream experiments or excluded?
  • Figure 4: Should also include survival data.
  • Figure 5: The author compare the Rnaset2 overexpressing tumors to both the parental and vector control tumors. However many of the results are only significant when compared to one of those groups and it is not consistent. This raises concerns about the validity of the finding given the small number of animals used.  Most likely power analysis would reveal that a much larger cohort of animals is needed to be confident in any observed effect.
  • The authors note an expansion of CD8 infiltrating cells in the Rnaset2 tumors and speculate that a T cell driven adaptive response is responsible for the ability to prevent tumor formation upon rechallenge. The authors should test this by preforming adoptive transfer experiments.  This would show if the T cell response is sufficient to confer the long term protective effect. 

Author Response

REVIEWER N.3

Comments and Suggestions for Authors

In this manuscript De Vito et al. analyze the role of Rnaset2 in modulating tumor growth using syngeneic mouse models. The authors first demonstrate that Rnaset2 over expression leads to reduced tumor volume in both a breast adenocarcinoma and a colon carcinoma model. They proceed to show that following rechallenge with parental cells, 75% of mice tested did not develop a new tumor. Finally, they conclude by examining the immunophenotype of the tumor associated macrophages, tumor infiltrating T-cells, and splenic cells.

This work is the first to study the effects of Rnaset2 in an in vivo immunocompetent setting and demonstrates that Rnaset2 expression can lead to an adaptive immune response which can protect from rechallenge, which is a novel finding that will be of interest to the field. However, there are many concerns about the methodology, rigor, and data interpretation that prevent me from recommending this manuscript for publication in its current form. Critically, the authors do not provide data on mouse survival or randomization of their study, and use unacceptably small numbers of mice without including any statistical justification.   

Figures do not look publication quality and all need reformatted

We are sorry to have produced low quality figures. In the new version of the manuscript we have therefore optimized all the figures which now present a high quality.

All figure legends need to include number of independent replicates, statistical tests used, and to indicate what p value markers of statistical significance correspond to.

Now the requested information has been included according to the referee’s suggestion.

More information about all mouse studies needs to be included. At a minimum: was power analysis preformed, was study randomized, and how was survival impacted.

Concerning information about all mouse studies, we estimated on our previous experience in in vivo studies, and used an ANOVA assay employing the three experimental groups, in which there is error type I a=0.05, and power analysis= 80%, with 6 animals per group. We did not do the randomization, as they were all cell lines that were injected. In this revision, we show a graph of the survival analysis.

More information is needed about how mice were selected for downstream experiments following tissue harvest.

As now indicated in figure legends in the revised version, we used six animals in group concerning experiments of tumor growth (both TS/A and C51) (Figure 2 and 3), ten animals in group for experiments with subsequent re-challenge (Figure 4), and six animals in group for IHC assays (Figure 5).

Figure 1: Lacks quantification of western blots.

The quantification data have been added as requested.

Figure 1: It is unclear which group Cl.1 belongs to

The figure has been reassembled to clarify the issue raised by the referee.

Figure 1: Probing for FLAG does not address the question of the amount of Rnaset2 overexpression. Need to probe using antibody that recognizes endogenous protein.  If unavailable, qRT PCR data would also be acceptable.

The results of a western blot analysis with a polyclonal anti-human RNASET2 on both parental C51 and TS/A cell lines protein have been included in the manuscript (Supplementary Figure 1). As shown in the figure, both murine cell lines displayed a negligible expression level of endogenous RNASET2 protein (the human ovarian cancer-derived OVCAR3 cell line is included in the blot as a positive control). This result is not due to the lack of cross-reactivity of our anti-human antibody to the murine RNASET2 protein, since the same antibody has been previously shown by our group to clearly detect murine RNASET2 in western blot analysis of several mouse adult tissues (Campomenosi et al. Characterization of RNASET2, the first human member of the Rh/T2/S family of glycoproteins. Arch Biochem Biophys. 2006 May 15;449(1-2):17-26. Epub2006 Mar 13. PubMed PMID: 16620762).

Figure 2: Needs to also include mouse survival data.

In the new version of manuscript, we have added in both figure 2, 3 and 4, mouse survival data.

Figure 2: While not required, images of mouse tumors would help manuscript

Regarding this request, we are sorry but we have not taken images of the various explanted tumors and we cannot recover them.

Figure 2B: Unclear if there is any statistical significance

We re-perform statistical analysis and did not show any statistical significance, i.e. p>0.05.

Figure 3: Legend is vague about which groups have 10 vs 6 mice

Now we have optimized legend and fixed it.

Figure 3b: Is there any statistical significance?

Fig 3B now has become Fig. 4C. And yes, we have statistical significance between C51 Rnaset2-injected mice and the two other groups.

Figure 3c: There are clearly outlier mice at the day 17 timepoint, if these mice are removed are the results still statistically significant? Also were these mice used for downstream experiments or excluded?

Concerning Figure 3C that now has become Figure 4C we only erase one mouse at day 20 post-injection because it had a large tumor beyond 1.5 cm3. The other tumor volumes were under this limit and then were all used for statistical evaluation.

Figure 4: Should also include survival data.

In the new version of manuscript, we have added mouse survival data in both figure 4, 3 and 2.

Figure 5: The author compare the Rnaset2 overexpressing tumors to both the parental and vector control tumors. However, many of the results are only significant when compared to one of those groups and it is not consistent. This raises concerns about the validity of the finding given the small number of animals used. Most likely power analysis would reveal that a much larger cohort of animals is needed to be confident in any observed effect.

Concerning this criticism from the reviewer, we could say that indeed, C51P and C51E control clones injected in mice showed a slight difference in some tumor infiltrate immune cells such as CD206+ M2 macrophages, granulocytes and CD31+ cells. Parental, untransfected C51 control cells have been added to the experimental plan in order to include a further negative control assay. In particular, we wanted to exclude unpredictable side-effects due to random integration in the mouse genome of empty pcDNA3 control plasmid in transfected cells. However, the in vivo growth kinetics showed a slight inhibition of C51 E tumor growth, albeit not significant, that may suggests a weak tumor suppressive effect in mice carrying empty vector versus C51P untransfected cells, and in this vision it might be related to either the different culture conditions in which these cells were grown in vitro before being injected in syngeneic mice (i.e., with or without the G418 selective agent for expression vector maintenance) or, alternatively, to the above-mentioned empty vector-induced genome integration effects. However, it should be noted that the slight differences between the two cell types (i.e. M2 and GR) and CD31 endothelial cells between the empty vector cells and C51 P are not statistically significant. We therefore reckon that, notwithstanding these discrepancies, our data provide quite convincing evidence about the role of the murine immune system in Rnaset2-mediated tumor suppression, also in light of the wide amount of literature data in support of such role for several members of the T2 RNase family. But further future investigations will be pursued that could allow to clarify this criticism, even if the interest of it is of moderate importance, in relation to no evident effect on the adaptive immune T cell compartment.

The authors note an expansion of CD8 infiltrating cells in the Rnaset2 tumors and speculate that a T cell driven adaptive response is responsible for the ability to prevent tumor formation upon re-challenge. The authors should test this by preforming adoptive transfer experiments.  This would show if the T cell response is sufficient to confer the long-term protective effect.

We agree with the comment of the referee. However, in this study we first wanted to investigate if there was a non-cell autonomous effect against the tumor in a murine BALB/c Rnaset2-based syngeneic in vivo assay, using two Rnaset2-overexpressing mouse C51 or TS/A syngeneic cells, and the in vivo behavior of these engineered tumor cells.

Although there was no in vitro evidence for inhibition of proliferation of TS/A Rnaset2-transfected cells, a trend for Rnaset2-mediated decrease of tumor growth rate in vivo was observed, thus we decided to focus on C51 colon carcinoma cells in which a strong and statistically significant difference was observed between mice injected with C51 FL Rnaset2 and empty-vector or C51 P cells. Furthermore, in this model we in deep analyzed in detail immune cell infiltrate and function of immune cells by different assays.

As shown on figure 3B, six animals were studied for tumor growth and there were significant differences in the rate of tumor growth in comparison to control injected mice and 33% (2 out of 6) of mice being able to reject primary engineered C51 FL Rnaset2 tumors. This assay was confirmed by another similar test involving 5 mice, obtaining similar results, with similar statistical significance and 40% (2 out of 5 mice) were able to reject again the primary engineered tumor. Then, in the third in vivo tumor growth assay, 10 animals were investigated and again 40% of mice were able to reject primary tumors, of which these 4 mice then underwent a re-challenge with C51 P cells and 75% of mice were able to reject completely the parental tumor, suggesting a Rnaset2-mediated T cell adaptive immune memory response, that was corroborated by our results on IHC showing great infiltrate of CD8+ T cells into C51 FL Rnaset2 tumors. Certainly, as suggested by the reviewer, an adoptive transfer test of CD8+ T immune cells from rejecting mice to newer mice receiving C51 challenge, would bring general confirmation of this suggested indication. But we think that our findings exposed in the revised manuscript contain a lot of new and interesting data, for example indicating a strong involvement of M1 and M2 macrophages, MDSCs and CD8+ T cells, as well as inhibition of CD31+ endothelial cells within Rnaset2+ tumor that allowed in vivo rejection capacity and that it will be necessary a new study to focus and confirm which subset of immune cells are involved in the rejection process.

Round 2

Reviewer 2 Report

  1. Figure 4C, additional mouse number of group 1 is needed for data presented in day 20.
  2. Figure 5, besides IHC/IF data, multi-color flow cytometry analysis are necessary.

  3. Figure 6, what is the meaning of the gating of R2? For analysis of immune cells in tumor, CD45 is necessary. 
  4. The difference of C51P and C51E should be mentioned and discussed in the manuscript.

  5. Label of x-axels in Figure 2C need to be modified. Please delete 23, 28, 32.

Author Response

Comments and Suggestions for Authors

In agreement with reviewer’s request, we double-checked English throughout the manuscript, making various corrections and improvements. We hope that the revised version is adequate and of good quality in writing.

-1- Figure 4C, additional mouse number of group 1 is needed for data presented in day 20.

Following suggestion from the reviewer we have checked with other additional mice for day 20 regarding C51 P injected animals, from other in vivo experiments in the C51 mouse model. And the new graph now showed indeed a statistical significance as indicated in the new figure 4C.

-2- Figure 5, besides IHC/IF data, multi-color flow cytometry analysis are necessary.

With reference to this request, it must be said that the biological material obtained from explanted tumors, in particular the engineered Rnaset2 ones, is rather small and the data we have produced are derived from many sacrificed animals. We therefore believe that the data presented in the revised manuscript are adequate to support our working hypotheses on the role of tumor suppressor of ribonuclease T2 in the mouse model, being our work the first to investigate this phenomenon in syngeneic tumor models. Therefore, the multi-color flow cytometry analysis experiments could be a way for new studies in the future to further investigate immune cell correlates and their functions related to the action of Rnaset2 molecule in our mouse model.

-3- Figure 6, what is the meaning of the gating of R2? For analysis of immune cells in tumor, CD45 is necessary. 

We sincerely apologize for the forgetfulness in the materials and methods of specifying the use of the murine anti-CD45 (or LCA) monoclonal antibody as a general staining or gate 1 analysis to identify all leukocytes from homogenate C51 tumor cells. We thank the reviewer because we had also forgotten to put the type of anti-F4/80 mAb used. The corrections are now highlighted in red in the new version.

We also modulate the sentence at the beginning of figure legend as follows:

“Figure 6. Cytometric flow assessment after in vitro tumor digestion has been performed to study percentage of tumor leucocyte (CD45+ (not shown) infiltration of CD86+ M1 and CD206+ M2 macrophage ratio in BALB/c mice (N=three 3 animals in group) injected with C51 P, C51 E and C51 FL Rnaset2 cell lines (panel A)…”

At the same in the M&M we have corrected the sentence as follows:

“Spleen cells or tumor infiltrating leucocytes were first incubated with an FcR blocking reagent (Miltenyi Biotec, Germany) for 10 min. Afterward, cells were labeled with anti-CD4 (clone GK1.5), anti-CD8 (clone 53-6.7), and anti-CD11b (clone M1/70) mAbs, all from eBioscience (San Diego, CA); anti-F4/80 (clone BM8) and anti-CD86 (clone GL-1) mAbs from BioLegend (San Diego, CA), and anti-CD206 (clone MR5D3) mAb from Bio Rad (Kidlington, UK). Briefly, after physical parameter setting (Forward and Side Scatter), and using anti-LCA CD45 (clone 30-F11) mAb from BioLegend mAb for tumor infiltrating leucocytes analysis, lymphocyte and myeloid populations were identified and then M1, M2, macrophages and T cells were evaluated by mAb-specific staining and assessed by FACS analysis using CELLQUEST or DIVA software (BD)…"

-4- The difference of C51P and C51E should be mentioned and discussed in the manuscript.

In agreement with the reviewer's request, we added new clarifying sentences in the discussion to try to explain this critical point, as follows:

“C51 P and C51 E control clones also showed a slight difference in in vivo tumor growth rates as well, although statistically not significant. Such difference might be related to either the different culture conditions in which these cells were grown in vitro before being injected in syngeneic mice (i.e., with or without the G418 selective agent for expression vector maintenance) or, alternatively, to empty vector-induced genome integration effects. It is moreover worth noting that the observed slight difference in the growth rate in vivo between C51 P and C51 E cells, showing a trend for a slightly more pronounced tumor growth in the former, was mirrored in most flow cytometry data shown in figure 7, where C51 P cells showed a more marked effect than C51 E cells with respect to Rnaset2-expressing cells in terms of several immunological-related parameters, which in some cases reached statistical significance. We thus speculate that the significant differences between C51 P and C51 E cells emerged in most single immunological assays carried out by cytometry are in keeping with the non-significant trend emerged in in vivo for the much complex process of growth rate assay. Thus, taken together we reckon that, notwithstanding a few discrepancies (such as the higher M1/M2 ratio in C51 P vs. C51 E tumors shown in figure 6) our data provide a strong evidence about the role of the murine immune system in Rnaset2-mediated tumor suppression, also in light of the wide amount of literature data supporting such role for several members of the T2 RNase family.”

-5- Label of x-axels in Figure 2C need to be modified. Please delete 23, 28, 32.

We agree with the reviewer's suggestion, and therefore now in the new figure 2 there are no longer the 3 additional dates, which led to confusion.

Reviewer 3 Report

The manuscript is much improved, and De Vito et al addressed all concerns that they were able to, in the revision time allotted. 

Author Response

Comments and Suggestions for Authors

The manuscript is much improved, and De Vito et al addressed all concerns that they were able to, in the revision time allotted. 

We thank reviewer 3 for his final thoughts and for appreciating all our efforts to improve the article.